# A Low-Cost, Integrated Immunization, Health, and Nutrition Intervention in Conflict Settings in Pakistan—The Impact on Zero-Dose Children and Polio Coverage

**DOI:** 10.3390/pathogens13030185

**Published:** 2024-02-20

**Authors:** Amira M. Khan, Imran Ahmed, Muhammad Jawwad, Muhammad Islam, Rehman Tahir, Saeed Anwar, Ahmed Ali Nauman, Zulfiqar A. Bhutta

**Affiliations:** 1Centre for Global Child Health, Hospital for Sick Children, Toronto, ON M5G OA4, Canada; amira.khan@sickkids.ca (A.M.K.); muhammad.islam@sickkids.ca (M.I.); 2Centre of Excellence in Women and Child Health, Aga Khan University, Karachi 74800, Pakistan; imran.ahmed@aku.edu (I.A.); muhammad.jawwad@aku.edu (M.J.); 3Trust for Vaccines and Immunization, Al Sehat Centre, Rafiqui Shaheed Road, Karachi 74350, Pakistan; rehman@tvi.org.pk; 4Prime Foundation, Peshawar Medical College, Warsak Road, Peshawar 25160, Pakistan; sanwar@piph.prime.edu.pk (S.A.); program@piph.prime.edu.pk (A.A.N.)

**Keywords:** polio eradication, zero dose, immunization, integrated interventions, conflict-affected, health systems

## Abstract

Pakistan is one of two countries globally still endemic for poliovirus. While increasing immunization coverage is a concern, providing equitable access to care is also a priority, especially for conflict-affected populations. Recognizing these challenges, *Naunehal*, an integrated model of maternal, newborn, and child health (MNCH), immunization, and nutrition services delivered through community mobilization, mobile outreach, and private-sector engagement was implemented in conflict-affected union councils (UCs) with high poliovirus transmission, including Kharotabad 1(Quetta, Balochistan) and Bakhmal Ahmedzai (Lakki Marwat, Khyber Pakhtunkhwa). A quasi-experimental pre–post-design was used to assess the impact of the interventions implemented between April 2021 and April 2022, with a baseline and an endline survey. For each of the intervention UCs, a separate, matched-control UC was identified. At endline, the proportion of fully immunized children increased significantly from 27.5% to 51.0% in intervention UCs with a difference-in-difference (DiD) estimate of 13.6%. The proportion of zero-dose children and non-recipients of routine immunization (NR-RI) children decreased from 31.6% to 0.9% and from 31.9% to 3.4%, respectively, with a significant decrease in the latter group. Scaling up and assessing the adoption and feasibility of integrated interventions to improve immunization coverage can inform policymakers of the viability of such services in such contexts.

## 1. Introduction

Pakistan remains one of only two countries globally with a continued endemic transmission of wild poliovirus [1]. While the national-level coverage of fully immunized children is around 76%, it remains much lower in certain regions with the data showing coverage at 37.6% and 42.8% in the provinces of Balochistan and Southern Khyber Pakhtunkhwa (KP), respectively, where being fully immunized refers to children (12–23 months old) who have completed routine immunization till Measles dose 1 (except the Rotavirus vaccine) as per the Expanded Program on Immunization (EPI) schedule [2]. Additionally, despite the slow progress in improving maternal, neonatal, and child health (MNCH) and nutrition indicators, Pakistan has the highest neonatal mortality rate globally at 39 deaths per 1000 live births and an under-five-years-old mortality rate of 63 deaths per 1000 live births [3]. Alarmingly, more than one-third of children under five years old in the country are stunted [4]. During 2020–2022, COVID-19 and its related mitigation strategies further challenged population health and health systems, with a sustained impact on both [5]. Although ensuring the provision of quality healthcare services is a concern, providing equitable access to care poses a challenge, especially for conflict-affected and marginalized populations. The shortage of healthcare staff and deliberate attacks on healthcare workers, especially those administering polio vaccinations, has hindered the distribution of MNCH services. Importantly, an escalation in violence and safety concerns have had a significant impact on community care-seeking behaviors, especially in KP and Balochistan [6]. Climate-related emergencies and food insecurity further exacerbate the situation. 

Given the multitude of issues and challenges in the context of polio and beyond, it is critical to implement innovative, low-cost strategies that can increase vaccination coverage, and improve accessibility to health services in insecure and conflict-affected contexts. The polio vaccination program in Pakistan, like most other health delivery strategies, is vertical and largely focuses on urban areas or refugee camps, with limited community buy-ins, and missed remote and internally scattered, displaced populations [7]. Additionally, the strategies for community mobilization and service delivery in such contexts are largely divided between implementing agencies, potentially increasing transaction costs while making the interventions quite complicated. 

Nearly all polio cases reported in Pakistan continue to be from seven polio-endemic districts in southern KP, where 1.1 million vaccine-eligible, under-five-years-old children are located [8]. Close to 50,000 of these children are missed in the government’s annual supplementary immunization activities (SIAs) as a result of refusals based on vaccine misconceptions, mistrust in health workers, and an emerging “polio fatigue” [1,9,10]. Researchers have linked community “polio fatigue” with weariness and cynicism regarding repeated polio immunization activities when the same community’s access to basic health and nutrition services remains limited [11]. Thus, it is not only critical to engage, educate, and mobilize communities, particularly community elders and religious leaders, but also to provide these populations with much-needed health and nutrition services. 

The disruption caused by COVID-19 halted vaccination drives and polio supplementary immunization activities in 2020, causing a further setback to polio vaccine coverage as well as routine vaccination rates, particularly in fragile, conflict-affected areas [12,13,14]. In such circumstances, implementing low-cost integrated strategies for providing health and immunization services where most required was critical. Recognizing these challenges and opportunities, *Naunehal*, an 18-month pilot project, was implemented in three high-risk union councils (UCs) of Pakistan, in 2021–2022. Previous work indicates that we have been working on community-based models for integrating polio, other immunizations, and MNCH interventions, and have been steadily working to simplify the approach and associated costs [15,16]. Thus, the *Naunehal* project adopted a low-cost, integrated model of MNCH, immunization, and nutrition services delivered through community mobilization, mobile outreach, and private-sector engagement strategies. The objective of this paper is to examine the coverage and utilization of the intervention and its impact on immunization coverage (including routine immunizations and oral polio vaccines (OPVs) administered during SIAs) in children under five years old, care-seeking practices, and nutrition behaviors, while exploring the impact on zero-dose children. 

## 2. Materials and Methods

### 2.1. Study Setting

*Naunehal* was implemented in three UCs of Pakistan. Kharotabad 1 (Quetta District, Balochistan Province) and Bhana Mari (Peshawar District, (KP) Province) were classified as super-high-risk UCs (SHRUCs) by the Pakistan Polio Eradication Programme, while UC Bakhmal Ahmedzai in district Lakki Marwat, KP, is an area that has experienced polio outbreaks in recent years. In addition to their high poliovirus transmission rates, the UCs also struggled with sub-optimal immunization coverage, maternal and children-under-five-years-old health, and nutrition indicators (Figure 1). Kharotabad-1, a peri-urban UC, and Bakhmal Ahmedzai, a rural UC, are also impacted by sectarian violence and regional insecurity, while Bhana Mari is in an area historically housing refugees. The three UCs were, thus, selected, given their complex contexts and high poliovirus transmission rates. 

### 2.2. Study Design

The project adopted a quasi-experimental pre–post design to assess the impacts of the interventions implemented between April 2021 and April 2022. The baseline survey was conducted in Feb–March 2021, while the endline survey was conducted in May 2022. The study protocol has been previously published by Ataullahjan et al. [17]. 

The three UCs were the intervention sites, and for each of these target UCs, a separate, matched-control UC of a comparable size, population, and location, and coverage indicators were identified, with propensity score matching and in consultation with local partners. The control UCs were Ward-11-A (Quetta District), Pahar Khel Thal (Lakki Marwat District), and Sheikh Junaidabad (Peshawar District), which, though always in the same district, never bordered the target UC. 

The project implemented three main strategies in the intervention UCs, including community mobilization, mobile health services, and the engagement of private healthcare providers (HCPs). Community engagement and mobilization included raising awareness via health information sessions that focused on routine childhood immunization; optimal infant and young child feeding (IYCF) practices; water, sanitation, and hygiene; optimal diarrhea management; and appropriate care seeking for both pregnant women and young children. A key component of the community mobilization was ongoing engagement with local leaders and religious scholars to gain their support and buy-ins for the project activities and health messages. 

The mobile health services included one team per UC, comprised of a female health worker, a vaccinator, and a facilitator, who visited pre-determined areas of the UC six days a week. The team provided basic health services, routine government-recommended immunization and health services, and IYCF counseling, targeted toward children under five years old and women of reproductive age (WRA). Additionally, the project also identified private HCPs in the intervention UCs to provide age-appropriate routine vaccinations to all under-five-years-old children visiting their clinics. The HCPs were trained in collaboration with the government’s District Health Office and EPI. The project also liaised with the EPI to supply the HCPs with vaccines free of cost on a regular, as-needed basis. During the 12-month intervention period, the government’s SIAs continued as scheduled both in the intervention and control UCs. 

Baseline, midline, and endline surveys were conducted using the 30 × 7 technique in the intervention and control UCs. Thirty clusters demarcated by the government’s polio program from each UC were randomly selected and 15 households with children under five years old were selected from each cluster. For each round of the survey, an updated listing of all households was generated for each cluster. The selection of households was performed through systematic random sampling. To achieve an optimal sample size, the target was to survey 450 households from each of the intervention and control UCs. The baseline that was to serve as the basis for targeting and tracking progress focused on household characteristics, immunization practices, health, nutrition, and care-seeking behaviors. As well, the survey examined the community’s awareness of and attitude toward COVID-19. The endline survey focused on the same indicators, but also assessed the household’s uptake and attitude toward the intervention. Immunization status was determined using the vaccination card. However, if the card was not available, parental recall was used to obtain the information. 

The survey inclusion criteria for a household were consent from the family, at least one child under 5 years of age in the household, and that the family had lived in that high-risk UC for at least 6 months (including the child in the household) before the day of the interview. However, the household was also eligible if the only child in the household was born in the last six months. Any household where a parent or caregiver was not available to answer questions related to the child’s health and immunization was excluded. The data collected were processed and analyzed at the Aga Khan University, Karachi. A consent form outlining the objectives of the study, how privacy concerns would be handled, and how the data would be used was shared with the participants. 

### 2.3. Statistical Analysis

Data related to demographics, clinical knowledge, attitude and practices, and MNCH-related indicators were compiled and analyzed using STATA version 18.0. To assess the independent effect of interventions, propensity score matching was employed to identify similar UCs based on the geographical location of the UC, population density, size of the children-under-five-years-old population, number of health facilities and health workers, and immunization coverage. Through this process, three matched UCs were also surveyed. Among the predictors, exact matching was enforced to achieve a balance for all the predictors between the intervention and non-intervention groups. Descriptive and inferential statistics were used to characterize the study sample and test hypotheses. Frequency and percentage were calculated for categorical variables and the mean (SD) was calculated for continuous variables. 

The mean and percentage point difference in the coverage between the baseline and endline was estimated using a generalized linear model with an identity link and binomial distribution.

We compared the change in prevalence of immunization, antenatal care, postnatal care, and newborn care indicators from the baseline to endline in the two arms using difference-in-difference (DiD) estimates. The DiD estimates were obtained from mixed linear regression models with an interaction term between the variables for arms (intervention vs. control) and time (endline vs. baseline).

The predictors of “zero-dose” and “non-recipient of routine immunization” (NR-RI) children were determined by the multivariable logistic regression after an initial univariable analysis. Variables significant at *p* < 0.2 in the univariable analysis were included in the fully adjusted model. The final model was constructed using a backward elimination, with variables being retained if *p* < 0.05. A *p*-value less than 0.05 was taken as significant. It is important to mention here that, for our analysis, “NR-RI” was defined as children who did not receive any routine vaccinations (BCG, OPV0, OPV1, Pentavalent (Penta) 1, Pneumococcal Conjugate Vaccine (PCV) 1, OPV2, Penta2, PCV2, OPV3, Inactivated Poliovirus Vaccine (IPV) 1, Penta3, PCV3, and Measles 1), while “zero dose” referred to children who did not receive any routine vaccinations and did not receive OPVs during SIAs. Thus, the NR-RI group was a sub-group of “zero-dose children”. On the other hand, “fully immunized” was defined as children who received age-appropriate doses for routine vaccines till Measles 1. 

The clustered nature of the data was accounted for by including each cluster as a random effect. Estimates were adjusted for the survey design and sampling weights by treating each UC as strata and clusters as primary sampling units.

### 2.4. Outcomes

The primary outcomes of this study were the coverage of OPVs and routine EPI vaccines, changes in the proportion of zero-dose children, changes in IYCF practices, and changes in care-seeking behaviors.

## 3. Results

At baseline, 1286 and 1277 households were visited in the intervention and control UCs, respectively, and the data for 4387 under-five-years-old children were collected. The baseline demographic characteristics of children under five years old at both sites were predominantly similar (Table 1). The male-to-female ratio for under-five-years-old children was almost equal, with a slight male preponderance. A significant majority of mothers (85.1%) in households surveyed at all intervention sites at baseline had no formal schooling—nearly 96% in UC Bakhmal Ahmedzai. Overall, for the intervention UC, 16.8% and 15.2% of the respondents belonged to the poorest and richest quintiles, respectively. However, this varied between UCs with the predominant population in Bhana Mari (intervention UC in Peshawar) belonging to the “rich” (44.1%) quintile, in Kharotabad-1 (intervention UC in Quetta) belonging to the “middle” quintile (46.8%), while in Bakhmal Ahmedzai (intervention UC in Lakki Marwat), most of the population belonged to the ”poorest” quintile (64.8%) (Table 1).

While nearly all intervention areas and control households in Lakki Marwat and Peshawar had an improved source of drinking water, the proportions were much lower in the intervention (32.1%) and control (46.1%) areas in Quetta. The improved source of sanitation in households was higher in the intervention and control UCs of Peshawar (99.8%, 99.8%), as compared to Quetta (87.5%, 96.9%) and Lakki Marwat (78.9%, 87.4%) (Table 1). As is apparent, the household characteristics and socioeconomic and education status were markedly better for Peshawar compared to the other districts. 

The main study outcomes also included IYCF practices and other health-related care-seeking behaviors related to women and children. Given their distinct determinants and relevance, the findings related to these outcomes and practices will be submitted and discussed in a separate manuscript. In this manuscript, we focused on vaccination-related practices and behaviors.

As shown in Table 2, the mobile health services data from the three sites show that there is a total of 30,768 beneficiaries for the 829 mobile health camp days, with the most camp days (319) in Kharotabad-1. An estimated 13,931 under-five-years-old children were provided services at the mobile health camps, with more than 18,000 immunization doses administered at all intervention sites. There were 73 vaccine refusals, of which 70 were for the oral polio vaccine, and the majority (n = 56) were in Kharotabad-1.

The immunization coverage component of the post-intervention endline survey for all three districts showed a proportion of fully immunized children that significantly increased from 27.5% to 51.0%, with a DiD estimate of 13.6% (4.5%, 22.8%), with a significant decrease in NR-RI children from 31.9% to 3.4% with a DiD estimate of −20.4% (−32.4, −8.5) (Table 3). The intervention UCs showed an overall decrease in zero-dose children from 31.6% to 0.9%; however, the decreasewas not significant given a similar decrease in the control areas (41.3% to 5.0%).

In comparison to the rural/peri-urban intervention UCs in the districts of Quetta and Lakki Marwat, Bhana Mari was an urban UC with a population with a higher educational and socioeconomic status (Table 1) and significantly higher immunization coverage at baseline (Appendix A, Appendix A). Given these differences, the authors performed an additional, separate analysis for the intervention and control UCs in Quetta and Lakki Marwat considering their similar contexts, conflict-affected environments, and sub-optimal health and immunization indicators (Appendix A, Appendix A). 

For these two districts, the proportion of fully immunized children increased significantly, from baseline to endline, from 20.2% to 43.2%, respectively, in intervention UCs with a DiD estimate % diff of 25.2 (17.1, 33.3) (*p*-value: <0.0001) (Table 4). The coverage for all routine vaccines from birth till 9 months of age increased significantly at endline. Similar to the all-site analysis, the proportion of zero-dose and NR-RI children in intervention UCs decreased from 46% to 1.1% and from 46.0% to 4.2%, respectively, with a significant decreased in the NR-RI category (Table 4). The UC-specific immunization coverage data are included in the Appendix A.

A detailed analysis of the characteristics of zero-dose children in all three districts showed that, at baseline, the gender distribution was almost equal with a predominant proportion (88.5%) having mothers with no formal schooling (Table 5). The majority (33.1%) belonged to the middle wealth quintile at baseline and the poor wealth quintile (44%) at endline. The commonest reasons provided by caregivers at baseline for not having these children vaccinated were a fear of side effects (19%) and a lack of faith in the immunization (17.4%). The multiple logistic regression analysis confirmed the odds of a zero-dose child having a mother with no formal education and belonging to the poorest quintile being significantly high (Appendix A, Appendix A). The same analysis for the NR-RI group showed that the odds were significantly high for these children to belong to the poorest and poor wealth quintiles (Appendix A, Appendix A). 

Importantly, an equity analysis showed that the intervention was successful in closing the equity gap for immunization coverage, especially for BCG, OPV 3, and Measles 1, demonstrating that an effective outreach program accompanied by community mobilization was instrumental in reducing equity-related barriers (Figure 2). 

## 4. Discussion

The results of our quasi-experimental study demonstrate that the integrated model tested for delivering immunization, health, and nutrition services in conflict-affected high-risk UCs in Pakistan is successful in significantly reducing the proportion of the NR-RI group of children (a subset of zero-dose children), reducing the proportion of zero-dose children, and significantly increasing the proportion of fully immunized children within the intervention areas via mobile health services and community engagement. The activity, in addition to the government-sponsored SIAs, also led to a significant increase in the coverage of all routine EPI vaccines at two conflict-affected intervention sites, with significantly positive impacts on vaccination card ownership and vaccination card retention. This relatively low-cost, simplified intervention was effective in reaching marginalized, at-risk populations and reducing key gaps in childhood immunization and a range of MNCH indicators. The program also included counseling for infant and young child feeding, as well as care seeking for common maternal and child health issues. As previously mentioned, these specific outcomes will be presented and discussed in a separate publication, while the vaccination-related outcomes and behaviors have been reported here.

As the global effort to eradicate polio gears up for the last mile, the focus is increasingly on targeting zero-dose children in the few, restricted areas with a virus transmission in the last two endemic countries: Pakistan and Afghanistan. The Polio Eradication Strategy 2022-2026 emphasizes the interruption of all poliovirus transmission chains by reaching zero-dose children in the seven high-risk, subnational, complex humanitarian areas, which include the Southern KP province in Pakistan [18]. Given the challenging task of reaching zero-dose children, the strategy also recommends an integrated approach to address both community hesitancy and low coverage, as well as to increase the long-term sustainability of the strategy. 

The results show that, while the proportion of zero-dose children is reduced in intervention UCs, a similar reduction also occurs in the control UCs. As per the definition, zero-dose children are those who have not received routine immunizations or OPVs during SIAs. In the post-COVID lockdown era, the Government of Pakistan actively implemented SIAs in Balochistan and KP, which targeted children under five years old for OPV administrations, and these would have been implemented across all intervention and control sites. The impact of the SIAs during the study intervention period explains the reduction in zero-dose children in both control and intervention UCs. However, the proportion of the NR-RI group did not reduce in the control UCs while intervention UCs saw a significant reduction, which could potentially be attributable to the mobile health services and community mobilization interventions of *Naunehal*. 

The impact of this project underscores the importance of using strategies that strive to “reach the unreached” populations in conflict-affected, underserved locations. This outreach model provides a set of interventions with immunizations integrated with basic maternal and child health services and nutritional counseling. Importantly, the health services being offered enhance the probability of the community interacting with the services and increase the likelihood of community members using immunization services, as have also been the experiences in Nigeria and Afghanistan [19,20]. Offering mobile immunization services in isolation would most likely not evoke the same uptake and response from the community, given the widespread vaccine skepticism and fatigue in these high-risk regions. The idea of a stand-alone “polio program” is presently viewed with suspicion and disdain by many communities; thus, the integration of these activities with other health services can work to build community trust [7]. Nevertheless, the skepticism and misinformation need to be counteracted by health education and counseling. 

India was declared polio free in 2014, and in the last stretch of the eradication efforts, the polio program also faced resistance grounded in “fear and fatigue” related to repeated OPV doses [21]. Moreover, it was realized that door-to-door vaccinations, although an effective strategy, did not help alleviate these fears as over-worked and under-resourced health workers were unable to address their concerns and questions. An added strength of the *Naunehal* model was a strong community engagement component, which not only included health information sessions with male and female community members and individual counseling sessions, but also a key focus on engaging with religious and community leaders as an initial step. The program, including mobile health services and private practitioner engagements, was not initiated till there was a complete buy-in from community elders, which in turn inspired trust and motivation from the community members, making them more open to awareness-raising and knowledge-sharing sessions. 

The model also reduced the risk of a missed opportunity for vaccination (MOV) where a vaccine-eligible child was aware of the health system but was not vaccinated for a variety of reasons, such as the failure of practitioners to screen him/her, vaccine shortages, or parental resistance [22]. A detailed vaccination coverage survey of all SHRUCs in Pakistan showed that nearly all SHRUCs had a high (>40%) proportion of children who were considered missed opportunities for simultaneous vaccinations (MOSVs) for IPV1, with the proportion being much higher in Balochistan [23]. The *Naunehal* mobile health services were designed in a way that the community was notified in advance of the visit date, and the service was scheduled to return to the same site again to coincide with the next vaccine doses, thus enhancing the convenience for the community and the uptake. Outreach services have proven to be a cost-effective strategy for delivering immunization services; however, the cost can escalate in fragile and remote settings, as has been noted [20,24,25]. Naunehal was implemented using a particularly cost-effective model with an optimal number of targeted, appropriately timed mobile health service visits using essential staff providing immunizations, healthcare services, and nutritional counseling. The added element of community mobilization guaranteed the community was aware and receptive, which ensured the efficient usage of the outreach program, thus minimizing costs. 

The strengths of the study include the assessment of the integrated strategy in a real-life, conflict-affected setting with a low-cost model. There exists a noticeable dearth of research assessing the delivery of immunization and health services in conflict-affected settings, and thus this study serves to fill the knowledge gap concerning the implementation and effectiveness of such models, especially in the context of zero-dose children and polio eradication [26,27]. Another strength of the study was the strong engagement of local communities and a close partnership with the government for vaccine provision, personnel training, and facility referrals, which increased the sustainability and cost-effectiveness of the model. 

A limitation of the study was that the model was implemented for a limited period and, thus, the sustainability and sustained effect of the program could not be established with the existing data. Additionally, the project encountered several challenges during its implementation. Firstly, navigating the program in an insecure context led to unpredictability where services and movements had to be occasionally suspended in certain areas, and plans adapted accordingly. Secondly, mobile health services and health information sessions were conducted during the challenging, initial waves of COVID-19. The teams followed protocols and worked in close coordination with government partners to avoid any breach of lockdowns or movement restrictions. Thirdly, private medical practitioner engagement could not be fully implemented as planned given, most notably, the lack of resources in private clinics to maintain the vaccine cold chain and the lack of incentives for busy practitioners to take on additional tasks.

## 5. Conclusions

As the efforts to eradicate polio intensify with experts striving to reach the remaining zero-dose and zero-routine immunization children, this study shows that an integrated model of delivering immunization, health, and nutrition services in combination with community mobilization in conflict-affected, high-risk UCs of Pakistan can reduce the proportion of the NR-RI group of children, a subset of zero-dose children, and increase coverage for all routine vaccines, including OPVs. The results of this pilot project have significant implications for polio eradication, suggesting new strategies for reaching marginalized, at-risk populations. It is advisable to scale up and evaluate this strategy at the population level in other conflict-affected contexts to examine the feasibility and effectiveness of increasing immunization coverage in high-transmission settings, especially for polio. 

## Figures and Tables

**Figure 1 pathogens-13-00185-f001:**
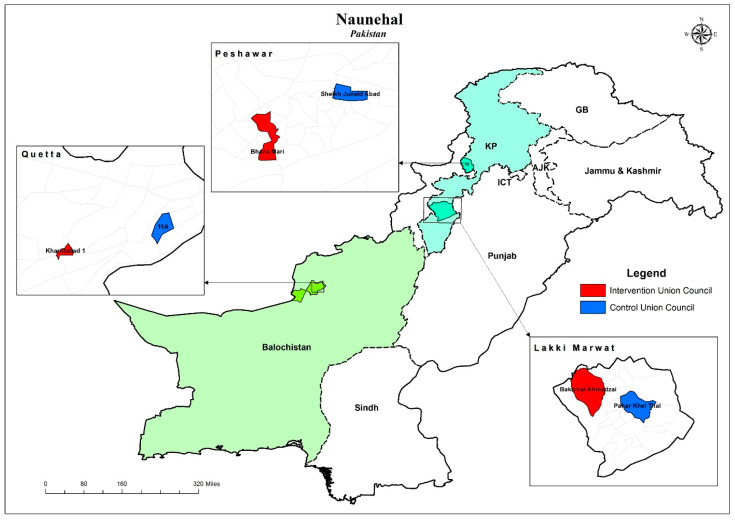
Map showing the control and intervention union councils.

**Figure 2 pathogens-13-00185-f002:**
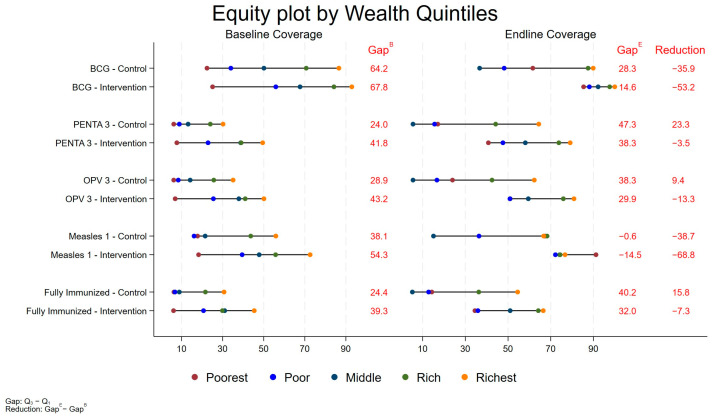
Equity Plot for Coverage of BCG, Penta3, OPV3, Measles1 and Fully Immunized Children (for all three districts).

**Table 1 pathogens-13-00185-t001:** Baseline sociodemographic characteristics of households surveyed and under-five-years-old children.

	Overall		Peshawar		Quetta		Lakki Marwat	
	Control	Intervention	Control	Intervention	Control	Intervention	Control	Intervention
Household Characteristics	N = 1277	N = 1286	N = 411	N = 401	N = 433	N = 446	N = 433	N = 439
Finished floor	773 (69.4)	696 (59.3)	404 (97.6)	390 (96.2)	344 (79.0)	253 (56.2)	25 (5.9)	53 (12.0)
Finished roof	1139 (87.9)	998 (76.2)	411 (100.0)	403 (99.2)	373 (84.5)	298 (63.0)	355 (82.3)	297 (65.7)
Finished walls	831 (69.0)	730 (59.7)	409 (98.9)	389 (96.1)	325 (71.8)	244 (52.1)	97 (23.2)	97 (20.9)
Solid fuel	443 (20.6)	464 (26.2)	5 (1.9)	14 (2.6)	6 (1.6)	11 (2.6)	432 (99.7)	439 (100.0)
Single room for sleeping	231 (12.3)	221 (13.9)	47 (14.6)	44 (10.0)	9 (2.3)	22 (5.3)	175 (38.8)	155 (33.9)
Improved source of drinking water	1053 (69.6)	972 (71.1)	410 (99.8)	406 (99.8)	210 (46.1)	141 (31.9)	433 (100.0)	425 (96.7)
Improved source of sanitation	1208 (95.7)	1147 (89.7)	410 (99.8)	406 (99.8)	423 (96.9)	392 (87.5)	375 (87.4)	349 (78.9)
Father’s Education								
No formal schooling	350 (33.3)	317 (28.8)	106 (26.3)	79 (24.2)	179 (42.1)	145 (35.5)	65 (16.2)	93 (24.0)
Primary	306 (23.5)	310 (24.0)	16 (6.0)	28 (5.9)	107 (24.6)	166 (37.2)	183 (42.2)	116 (27.6)
Secondary	387 (27.5)	389 (27.5)	142 (35.0)	158 (36.2)	99 (22.4)	96 (19.6)	146 (33.0)	135 (28.5)
Intermediate or above	237 (15.8)	278 (19.7)	147 (32.7)	142 (33.6)	51 (10.9)	41 (7.8)	39 (8.6)	95 (19.9)
Mother’s Education								
No formal schooling	1091 (86.1)	1109 (85.1)	271 (65.1)	285 (71.0)	396 (91.0)	406 (90.8)	424 (98.3)	418 (95.7)
Primary	35 (3.9)	58 (5.0)	10 (2.8)	17 (4.6)	23 (5.6)	29 (6.6)	2 (0.4)	12 (2.7)
Secondary	71 (5.2)	69 (5.5)	55 (14.5)	56 (13.0)	14 (2.9)	9 (1.9)	2 (0.3)	4 (0.8)
Intermediate or above	83 (4.8)	58 (4.4)	75 (17.6)	49 (11.4)	3 (0.6)	4 (0.7)	5 (1.0)	5 (0.8)
Wealth Index (Quintiles)								
Poorest	223 (10.1)	292 (16.8)	0 (0.0)	6 (1.1)	1 (0.2)	7 (1.6)	222 (51.9)	279 (64.8)
Poor	253 (16.3)	262 (19.5)	6 (2.1)	14 (2.6)	58 (13.3)	121 (28.8)	189 (43.0)	127 (27.8)
Middle	218 (27.1)	297 (26.1)	17 (5.4)	52 (14.5)	181 (44.1)	212 (46.8)	20 (4.7)	33 (7.4)
Rich	260 (25.2)	255 (22.4)	120 (32.7)	174 (44.1)	139 (30.5)	81 (17.4)	1 (0.2)	0 (0.0)
Richest	326 (21.2)	188 (15.2)	268 (59.8)	161 (37.6)	57 (11.8)	27 (5.4)	1 (0.2)	0 (0.0)
Children Under Five Years Old	N = 2175	N = 2212	N = 542	N = 574	N = 992	N = 946	N = 641	N = 692
Gender								
Male	1133 (51.5)	1124 (50.9)	288 (52.4)	273 (48.1)	502 (50.9)	498 (52.3)	343 (53.4)	353 (51.0)
Female	1042 (48.5)	1088 (49.1)	254 (47.6)	301 (51.9)	490 (49.1)	448 (47.7)	298 (46.6)	339 (49.0)
Age (Months)								
0–5	178 (7.9)	193 (8.2)	52 (9.3)	52 (8.3)	76 (7.6)	70 (7.2)	50 (7.8)	71 (10.2)
6–11	187 (8.9)	204 (10.0)	61 (12.5)	77 (14.1)	80 (8.4)	85 (9.5)	46 (7.4)	42 (5.8)
12–23	405 (18.7)	380 (17.2)	115 (20.6)	126 (21.0)	181 (18.6)	156 (16.6)	109 (17.0)	98 (13.7)
24–59	1405 (64.4)	1435 (64.7)	314 (57.7)	319 (56.5)	655 (65.3)	635 (66.8)	436 (67.9)	481 (70.3)

**Table 2 pathogens-13-00185-t002:** Mobile health services data.

	Overall	Bhana Mari	Bakhmal Ahmedzai	Kharotabad
Mobile health camp days	829	252	258	319
Total beneficiaries	30,768	7524	7225	16,019
Women of a Reproductive Age (WRA) in camps	10,572 (34.4%)	1851 (24.6%)	3039 (42.1%)	5682 (35.5%)
Pregnant women in camps	4416 (41.8%)	661 (35.7%)	824 (27.1%)	2931 (51.6%)
Children under five years old at camps	13,931 (45.3%)	5550 (73.8%)	4157 (57.5%)	4224 (26.4%)
Children under 5 years old provided with routine immunization	5193 (37.3%)	962 (17.3%)	3082 (74.1%)	1149 (27.2%)
Children under 5 years old provided with an OPV	7684 (55.2%)	4310 (77.7%)	1402 (33.7%)	1972 (46.7%)
Total number of immunization doses administered				
BCG	890 (6.4%)	74 (1.3%)	648 (15.6%)	168 (4.0%)
OPV	3721 (26.7%)	532 (9.6%)	2739 (65.9%)	450 (10.7%)
Penta	3541 (25.4%)	471 (8.5%)	2654 (63.8%)	416 (9.8%)
PCV	3526 (25.3%)	466 (8.4%)	2645 (63.6%)	415 (9.8%)
IPV	1344 (9.6%)	211 (3.8%)	1031 (24.8%)	102 (2.4%)
Rota	2554 (18.3%)	293 (5.3%)	1941 (46.7%)	320 (7.6%)
Measles	2501 (18.0%)	421 (7.6%)	1313 (31.6%)	767 (18.2%)
Vaccine Refusals				
Routine immunization	3 (0.0%)	2 (0.0%)	0 (0.0%)	1 (0.0%)
OPV	70 (0.5%)	7 (0.1%)	7 (0.2%)	56 (1.3%)

**Table 3 pathogens-13-00185-t003:** Routine immunization coverage at baseline and endline for children under three years old in intervention and control UCs in all three districts.

	Baseline		Endline		Control—% Diff (95% CI)	Intervention—% Diff (95% CI)	DID Estimate—% Diff (95% CI)	DID Estimate—*p*-Value
	Control	Intervention	Control	Intervention				
	N = 1122	N = 1160	N = 961	N = 1000				
Immunization Status								
Fully immunized	210 (15.9)	305 (27.5)	282 (25.8)	478 (51.0)	9.9 (3.2, 16.6)	23.6 (17.2, 29.9)	13.6 (4.5, 22.8)	0.0038
Partially immunized	463 (42.5)	430 (40.6)	393 (40.7)	473 (45.6)	−1.8 (−9.2, 5.6)	5.0 (−2.6, 12.6)	6.8 (−3.7, 17.3)	0.2039
Non-recipients of routine immunizations	449 (41.6)	425 (31.9)	286 (33.5)	49 (3.4)	−8.1 (−18.3, 2.1)	−28.6 (−34.9, −22.2)	−20.4 (−32.4, −8.5)	0.0009
Zero dose	447 (41.3)	422 (31.6)	47 (5.0)	13 (0.9)	−36.3 (−43.6, −29.0)	−30.8 (−37.4, −24.1)	5.5 (−4.3, 15.4)	0.2676
Ever had a vaccination card	537 (44.5)	640 (60.6)	616 (57.3)	912 (93.2)	12.8 (2.2, 23.4)	32.6 (25.6, 39.6)	19.8 (7.2, 32.4)	0.0022
At Birth								
	N = 1122	N = 1160	N = 961	N = 1000				
BCG	665 (57.8)	722 (66.9)	664 (65.4)	919 (92.9)	7.6 (−2.8, 18.0)	26.0 (19.4, 32.5)	18.3 (6.1, 30.6)	0.0034
OPV0	631 (53.8)	708 (65.6)	576 (53.5)	873 (89.0)	−0.4 (−11.2, 10.4)	23.4 (16.7, 30.0)	23.8 (11.1, 36.4)	0.0003
At 6 Weeks								
	N = 1093	N = 1134	N = 942	N = 980				
OPV1	335 (26.6)	456 (41.8)	544 (50.9)	799 (81.4)	24.4 (14.5, 34.3)	39.6 (32.4, 46.8)	15.2 (3.1, 27.4)	0.0141
Penta 1	559 (47.2)	557 (51.8)	562 (52.1)	850 (86.4)	4.8 (−5.4, 15.0)	34.6 (26.9, 42.4)	29.8 (17.1, 42.5)	<0.0001
PCV 1	538 (45.2)	545 (50.5)	563 (52.2)	845 (85.9)	7.0 (−3.2, 17.2)	35.4 (27.5, 43.3)	28.4 (15.6, 41.3)	<0.0001
Rota Virus 1	531 (43.9)	566 (52.8)	558 (51.8)	827 (83.7)	7.8 (−2.3, 18.0)	30.9 (23.0, 38.8)	23.0 (10.3, 35.8)	0.0005
At 10 Weeks								
	N = 1056	N = 1111	N = 909	N = 956				
OPV 2	259 (21.6)	403 (38.7)	378 (36.3)	642 (69.2)	14.7 (6.2, 23.2)	30.5 (23.5, 37.6)	15.9 (5.0, 26.8)	0.0046
Penta 2	295 (23.8)	430 (40.8)	491 (47.4)	710 (74.7)	23.6 (14.4, 32.8)	33.9 (26.2, 41.7)	10.3 (−1.6, 22.3)	0.0891
PCV 2	292 (23.6)	425 (40.4)	488 (47.2)	703 (73.9)	23.6 (14.5, 32.8)	33.5 (25.7, 41.3)	9.8 (−2.1, 21.8)	0.1056
Rota Virus 2	288 (23.3)	412 (39.2)	483 (46.6)	700 (73.3)	23.3 (14.2, 32.3)	34.1 (26.5, 41.7)	10.9 (−0.9, 22.6)	0.0695
At 14 Weeks								
	N = 1024	N = 1080	N = 882	N = 921				
OPV 3	229 (19.7)	343 (34.0)	315 (30.3)	563 (63.7)	10.6 (2.6, 18.7)	29.7 (22.8, 36.6)	19.1 (8.5, 29.6)	0.0004
Penta 3	209 (18.1)	338 (33.4)	309 (30.3)	530 (60.6)	12.2 (4.0, 20.4)	27.2 (19.9, 34.6)	15.0 (4.1, 25.9)	0.0071
PCV 3	209 (18.1)	330 (32.5)	298 (29.3)	527 (60.4)	11.2 (3.2, 19.2)	27.9 (20.8, 35.0)	16.7 (6.1, 27.3)	0.0022
IPV 1	432 (36.3)	460 (44.5)	448 (44.4)	654 (71.6)	8.1 (−1.3, 17.5)	27.1 (19.2, 34.9)	19.0 (6.8, 31.2)	0.0024
At 9 Months								
	N = 853	N = 875	N = 763	N = 767				
Measles 1	331 (32.4)	397 (47.7)	420 (47.5)	591 (76.0)	15.1 (5.9, 24.3)	28.4 (20.9, 35.8)	13.3 (1.5, 25.1)	0.0269
At 15 Months								
	N = 639	N = 662	N = 598	N = 565				
Measles 2	164 (19.8)	230 (35.1)	247 (37.7)	307 (54.6)	17.9 (9.1, 26.7)	19.5 (11.3, 27.6)	1.6 (−10.3, 13.4)	0.7963

**Table 4 pathogens-13-00185-t004:** Routine immunization coverage at baseline and endline for children under three years old in intervention and control UCs in the districts of Quetta and Lakki Marwat.

	Baseline		Endline		Control—% Diff (95% CI)	Intervention—% Diff (95% CI)	DID Estimate—% Diff (95% CI)	DID Estimate—*p*-Value
	Control	Intervention	Control	Intervention				
	N = 789	N = 810	N = 597	N = 700				
Immunization Status								
Fully immunized	74 (9.6)	144 (20.2)	60 (7.4)	270 (43.2)	−2.1 (−6.0, 1.8)	23.0 (15.8, 30.2)	25.2 (17.1, 33.3)	<0.0001
Partially immunized	275 (38.7)	250 (33.7)	257 (42.7)	384 (52.6)	4.0 (−5.6, 13.6)	18.9 (10.3, 27.4)	14.8 (2.1, 27.5)	0.0225
Non-recipients of routine immunizations	440 (51.8)	416 (46.0)	280 (49.9)	46 (4.2)	−1.9 (−13.6, 9.8)	−41.9 (−49.8, −34.0)	−40.0 (−54.0, −26.0)	<0.0001
Zero dose	438 (51.4)	416 (46.0)	42 (6.8)	12 (1.1)	−44.6 (−52.6, −36.5)	−45.0 (−53.3, −36.7)	−0.4 (−11.8, 11.0)	0.9432
Ever had a vaccination card	258 (35.7)	335 (47.1)	265 (37.3)	632 (93.2)	1.6 (−9.4, 12.6)	46.1 (37.8, 54.4)	44.5 (30.9, 58.1)	<0.0001
At Birth								
	N = 789	N = 810	N = 597	N = 700				
BCG	342 (47.6)	390 (53.7)	309 (48.9)	622 (90.5)	1.2 (−10.6, 13.1)	36.8 (28.5, 45.2)	35.6 (21.3, 49.9)	<0.0001
OPV0	312 (42.8)	369 (50.6)	231 (32.0)	590 (87.1)	−10.8 (−21.6, 0.0)	36.5 (28.4, 44.5)	47.2 (34.0, 60.5)	<0.0001
At 6 Weeks								
	N = 766	N = 786	N = 589	N = 683				
OPV1	152 (18.3)	253 (34.8)	226 (31.3)	528 (77.8)	13.0 (3.4, 22.7)	43.0 (34.2, 51.7)	29.9 (17.1, 42.8)	<0.0001
Penta 1	258 (35.5)	288 (40.1)	243 (32.9)	580 (85.1)	−2.6 (−13.1, 7.8)	45.0 (35.4, 54.6)	47.6 (33.6, 61.6)	<0.0001
PCV 1	242 (33.2)	281 (39.1)	243 (32.9)	578 (84.7)	−0.3 (−10.6, 10.0)	45.6 (36.0, 55.3)	45.9 (31.9, 59.9)	<0.0001
Rota Virus 1	228 (31.2)	277 (38.6)	243 (32.9)	565 (82.1)	1.7 (−8.5, 11.9)	43.5 (33.5, 53.5)	41.8 (27.7, 55.9)	<0.0001
At 10 Weeks								
	N = 744	N = 768	N = 572	N = 663				
OPV 2	100 (13.8)	218 (32.2)	106 (14.4)	393 (62.2)	0.5 (−5.5, 6.5)	30.0 (21.5, 38.5)	29.5 (19.2, 39.8)	<0.0001
Penta 2	129 (15.8)	233 (33.8)	210 (29.9)	456 (69.6)	14.1 (4.7, 23.5)	35.8 (26.3, 45.3)	21.7 (8.4, 35.0)	0.0015
PCV 2	126 (15.5)	229 (33.4)	208 (29.7)	453 (69.0)	14.2 (4.8, 23.6)	35.6 (26.0, 45.1)	21.4 (8.1, 34.6)	0.0017
Rota Virus 2	125 (15.5)	225 (32.8)	209 (29.8)	449 (68.0)	14.3 (4.9, 23.8)	35.2 (25.6, 44.7)	20.8 (7.6, 34.1)	0.0023
At 14 Weeks								
	N = 723	N = 743	N = 559	N = 637				
OPV 3	84 (12.2)	174 (27.1)	80 (9.8)	328 (55.1)	−2.3 (−7.4, 2.7)	27.9 (19.3, 36.5)	30.2 (20.4, 40.1)	<0.0001
Penta 3	83 (12.1)	174 (27.1)	65 (8.6)	298 (51.6)	−3.6 (−8.3, 1.2)	24.4 (15.5, 33.4)	28.0 (18.0, 38.0)	<0.0001
PCV 3	83 (12.1)	166 (25.8)	65 (8.6)	296 (51.4)	−3.5 (−8.3, 1.3)	25.6 (17.0, 34.2)	29.1 (19.4, 38.8)	<0.0001
IPV 1	167 (22.8)	223 (32.9)	208 (30.0)	440 (70.2)	7.2 (−2.2, 16.7)	37.3 (28.5, 46.1)	30.0 (17.3, 42.8)	<0.0001
At 9 Months								
	N = 605	N = 619	N = 504	N = 534				
Measles 1	132 (20.7)	206 (36.8)	215 (34.0)	408 (75.0)	13.3 (3.9, 22.8)	38.2 (29.7, 46.7)	24.9 (12.2, 37.5)	0.0001
At 15 Months								
	N = 437	N = 458	N = 403	N = 388				
Measles 2	51 (10.6)	105 (25.1)	108 (24.6)	184 (47.9)	14.0 (4.8, 23.2)	22.8 (13.3, 32.3)	8.8 (−4.3, 21.9)	0.1862

**Table 5 pathogens-13-00185-t005:** Characteristics of zero-dose children in all three districts.

	Overall			
	Baseline	Endline	% Diff (95% CI)	*p*-Value
	N = 869	N = 60		
Gender				
Male	440 (49.4)	29 (47.7)	−1.7 (−13.2, 9.7)	0.7675
Female	429 (50.6)	31 (52.3)	1.7 (−9.7, 13.2)	0.7675
Maternal Education				
No formal schooling	790 (88.5)	58 (97.4)	8.9 (3.5, 14.3)	0.0015
Primary	55 (9.3)	1 (0.7)	−8.6 (−12.1, −5.0)	<0.0001
Secondary	14 (1.4)	0 (0.0)		
Intermediate or above	10 (0.8)	1 (2.0)	1.2 (−2.9, 5.2)	0.5729
Wealth Quintile				
Poorest	285 (20.8)	20 (23.5)	2.7 (−14.2, 19.6)	0.7508
Poor	256 (25.6)	26 (44.0)	18.4 (3.1, 33.6)	0.0186
Middle	202 (33.1)	8 (21.1)	−12.0 (−29.0, 5.0)	0.1662
Rich	95 (15.9)	4 (7.8)	−8.1 (−19.0, 2.8)	0.1435
Richest	31 (4.6)	2 (3.6)	−1.0 (−8.5, 6.5)	0.7930
Reason for Not Receiving Immunization				
Place of immunization too far	95 (15.7)	4 (7.1)	−8.5 (−16.9, −0.1)	0.0462
Inconvenient/unknown timing/long wait	47 (6.0)	4 (3.5)	−2.5 (−6.7, 1.8)	0.2492
Parent/caretaker busy	58 (6.2)	4 (8.2)	2.0 (−9.1, 13.1)	0.7231
Child not well	54 (7.1)	2 (2.3)	−4.9 (−9.3, −0.5)	0.0305
Rumors	127 (12.3)	6 (9.2)	−3.1 (−13.3, 7.1)	0.5452
No faith in immunization	141 (17.4)	18 (31.2)	13.9 (−3.9, 31.7)	0.1256
Fear of side effects	205 (19.0)	0 (0.0)		
Vaccinator/vaccine not available	69 (5.6)	1 (3.4)	−2.2 (−9.5, 5.1)	0.5477
Others	5 (0.9)			
Do not know	39 (6.7)	4 (5.9)	−0.8 (−8.0, 6.3)	0.8221

## Data Availability

Data are available on request due to restrictions.

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
