# Peer review of "A Low-Cost, Integrated Immunization, Health, and Nutrition Intervention in Conflict Settings in Pakistan—The Impact on Zero-Dose Children and Polio Coverage"

_pathogens, 2024, doi:10.3390/pathogens13030185_

Round 1

Reviewer 1 Report

Comments and Suggestions for Authors

Reviewer Comments

Manuscript number: pathogens-2850642

Title: A Low-Cost Integrated Immunization, Health, and Nutrition Intervention in Conflict Settings of Pakistan - the Impact on Zero-Dose Children and Polio Coverage

Summary

As polio eradication nears actualization, interruption of both wild and vaccine derived poliovirus transmission globally becomes crucial. However, interruption of wild poliovirus transmission in Afghanistan and Pakistan (the two remaining endemic regions globally) is a priority. The authors describe a quasi-experimental study done in Pakistan in which they implemented an integrated model for delivering immunization, health, and nutritional services in conflict-affected high-risk union councils. They show the effectiveness of their integrated model in reducing zero-dose children and increasing the proportion of vaccinated children in the study population.

Recommendation

The authors have done well, and their study deserves to be published as it adds to the body of knowledge on strategies for “reaching the unreached” with health and nutritional services, especially in conflict-affected high-risk areas of the world. The study deserves revision.

General Comments

1.     Please be consistent. You use “not-vaccinated” and “not-immunized” interchangeably. Please correct all mentions of “not-immunized” to “not-vaccinated”

2.     Please mention the supplementary files in the manuscript.

3.     Page 1, Line 15: Please define MNCH on first mention.

4.     Page 6, Line 198: Please consider changing ‘servicdes’ to ‘services’.

5.     Table 4 shows that proportion of zero-dose children dropped in both the control and invention. Please speak to this in the discussion.

6.     Table 4 also shows that while proportion of zero-dose children dropped in the control population, the same did not really happen with the not-vaccinated children. Please speak to this in the discussion.

7.     Page 7, Lines 218-223: Based on Tables 6 and 7, please consider highlighting the odds of a child that is neither a “zero-doser” or “not-vaccinated” fitting that profile.

8.     Page 10, Line 276: Please consider changing ‘of eradication’ to ‘of the eradication’.

Reviewer 2 Report

Comments and Suggestions for Authors

The paper presents the results of an integrated Health program designed for application in remote and conflict affected areas of Pakistan. The work is well designed, well written and is of significant importance. In fact, the long lasting WHO polio eradication program that is ongoing since 1988 has been able to eradicate circulation of wild poliovirus from all but two countries worldwide. In fact, the most relevant areas of endemic wild strain poliovirus circulation are geographically the ones in the study of areas in the vicinity in Afghanistan. Since no non-human reservoirs of poliovirus are known, WHO had long hopped to have wild poliovirus circulation eradicated which has justified the over 20 billion US dollars already spent in the effort. However, worldwide efforts  have been hampered by the difficulty of full immunization in populations suffering from poverty, with difficult access to health care, located in conflict prone geographical areas and with low levels of education. All these problems are present at the same time in still endemic polio regions of Afghanistan and Pakistan, making it particular relevant any efforts into overcoming immunization resistance in these populations. 

Despite being well designed and well written, the paper has a few issues that should be overcome:

1) Major problems

    • in section 2.4 outcomes were defined for vaccination (OPV and routine vaccines), changes in zero-dose vaccination children and in infant feeding practices and care seeking behaviors. However only vaccine related outcomes are described in results. Either all outcomes are properly presented or they should clearly stated to have been or will be presented elsewhere. This is even more important as the non-vaccination related outcomes are actually discussed in lines 251-253. 

    • Analysis of outcome data from one of the three populations studied was excluded based on the population characteristics found. This seems to be a bad decision that ends up damaging the design of the study. All data should be presented. Analysis of data could then be done separately and validation of the intervention could be done in one or the two population settings. By excluding the data obtained from one population no data driven conclusions can be drawn, and questions are raised regarding the validity of the papers conclusion in all studied populations.

    • The logistic regression analysis made results in very little added information and may actually present an effort to draw conclusions on the importance of social, demographic and ethnic factors to explain the occurrence of zero-dose children. It results in too many tables, renders the paper much more difficult to read and results in very little information. Thus, it could be positive to describe these results in a brief manner and use the tables as supplementary data not included in print. It could even be argued that the language speaking population segregation in this analysis may present ethical concerns as it stratifies risk on the basis of cultural differences, regardless of education and personal effort. For this reason, care should be taken into not giving too much strength to this kind of analysis in an otherwise very interesting paper.

    • The most important issue regarding this paper is the interpretation of the results regarding zero-dose children. As can be seen in table 4, despite the fact that the intervention population showed a dramatic reduction in zero dose children (from 46% before intervention to 1.1% after the intervention) the same occurred in the control population (from 51.4% at base line to 6.8% at the endline). This resulted in a clearly non-significant p value (0.9432) indicating that some other factor besides de intervention program could be at play. Despite this authors clearly state in lines 210-211. Although results from not-vaccinated children (drop from 46.5% to 4.2% in the intervention groups versus 52.2% to 49.9% in control groups) resulted in a significant effect (p<0.0001) attributable to the intervention, this cannot be extrapolated to the zero-dose children results. Despite this, in the abstract (lines 23-24) the difference in zero-dose children found in the intervention group is described as resulting from the intervention, which is not supported by the data. This must be changed . The same occurs in the discussion (lines 244-245).

2) Minor issues:

    • table 3, third line. The description is missing “Age” in “Women of Reproductive Age (WRA)”

    • line 201. There is a speling error. “servicdes” has an extra d.
